# Variations in Children's screen time behaviors by weight status among a mostly disadvantaged population

Paul Son[1], Yuxin Nie[1], Qiaoyin Tan[1], Pengfei Yang[1], Peyton Murray[2],
Amanda E. Staiano[2], Fahui Wang[3], Gang Hu[2], Stewart Gordon[2,4], Senlin Chen[1]*

1 School of Kinesiology, Louisiana State University, Baton Rouge, Louisiana, United States of America,
2 Pennington Biomedical Research Center, Baton Rouge, Louisiana, United States of America,
3 Department of Geography & Anthropology, Louisiana State University, Baton Rouge, Louisiana, United States of America, 4 Louisiana Healthcare Connections, Baton Rouge, Louisiana, United States of America

* senlinchen@lsu.edu

## Abstract

The purpose of this survey study was to comprehensively examine if various types of screen time differed between children with overweight or obesity (OWOB; ≥ 85th BMI percentile) vs. normal weight (NW: < 85th BMI percentile) in a largely socioeconomically disadvantaged population. 739 parent proxies of children aged 5–11 years ($M = 9.27$, $SD = 1.49$) mostly enrolled in Medicaid (83.9%) in a United States (US) southern state responded to an online questionnaire called the *Movement Behavior Questionnaire – Child* (MBQ-C; open version). Eight items of the MBQ-C separately measured passive and interactive screen time on weekdays and weekend days. The survey also gathered parent-reported child weight and height, along with their sociodemographic characteristics. Compared to children with NW ($M = 294.5 \pm 7.2$ min/day), children with OWOB ($M = 364.3 \pm 10.6$ min/day) reported greater amount of total screen time ($M_{Diff} = 69.8$ min/day, 95% $CI = [45.0–94.6]$, $p < 0.001$). Of the sample, 24.9% met the sedentary screen time guidelines of no more than 2 hours/day favoring children with NW ($OR = 0.59$, 95% $CI$ [0.39, 0.86], $p = 0.008$). The difference of screen time between weight-status groups was greatest in passive screen time, particularly on weekdays ($M_{Diff} = 63.2$ min/day, 95% $CI = 47.6–78.7$, $p < 0.001$). Demographic factors did not significantly moderate the relationship between screen time and weight status. Most of the disadvantaged children failed to meet the screen time guidelines. Children with OWOB reported higher screen time, particularly passive screen time on weekdays. These findings suggest the need for tailored interventions to not only curb overall screen time but also mitigate specific types of screen time behaviors on specific days for children with OWOB (i.e., passive screen use on weekdays).

**Data availability statement:** The dataset presented in this article are not readily available unless a data-sharing agreement is established between the research team and the request individuals or party, upon the Louisiana State University IRB's review and approval due to the ethical restrictions with the vulnerable Medicaid population involved in the research. Requests to access the dataset should be directed to Louisiana State University Institutional Review Board (irb@lsu.edu).

**Funding:** This study was supported by the Public University Partnership Program at the Louisiana Department of Health, Bureau of Health Services Financing (LDH-PUPPAM230210), Louisiana State University Provost's Fund, and Louisiana State University Foundation/Our Lady of the Lake Health. Dr. Staiano and Dr. Hu were partially supported by grants from the National Institute of Diabetes and Digestive and Kidney Diseases (P30DK072476) and the National Institute of General Medical Sciences (U54GM104940).

**Competing interests:** The authors have declared that no competing interests exist.

## Introduction

Child health predicts adult health, with early risks tracking across the life course. For example, children with obesity are five times more likely to become adults with obesity, and they are at elevated risks for diabetes and cardiovascular disease [1,2]. Approximately 1 in 5 U.S. children and adolescents have obesity, illustrating the urgent need for effective interventions [3,4]. Studies have reported that screen time is a risk factor associated with obesity and chronic diseases across populations [5–10]. Restricting recreational screen use to no more than 2 hours per day is a recommended global consensus by the World Health Organization (WHO) as well as national health authorities in Canada and Australia [11–13]. However, most children aged 8–12 years spend almost 5 hours per day in front of an electronic screen, and tweens and teens' screen time exceeded 7 hours daily [14]. Compared to children with normal weight (NW), those with overweight or obesity (OWOB) have shown greater amount of screen time [15,16]. Children with OWOB are 33% less likely to adhere to the screen-time guidelines compared to children with NW [17]. Only 29.7% of children with obesity met the screen-time recommendation, compared with 38.6% of normal weight children [18].

Despite the known discrepancy in screen time by weight status, most existing evidence has examined screen time as one overarching behavior [19]. Rarely has prior research examined the various types of screen time behaviors and how they would differ by weight status. Screen time is not a one-dimensional construct. Instead, it may vary by engagement type (passive vs. interactive) and posture related-context (typically sedentary vs. standing), and children's screen exposure levels (type and duration) may differ by the day of week (e.g., weekdays vs. weekend days), which have not been comprehensively investigated in prior research, especially in populations from lower socioeconomic status backgrounds [20]. According to the *Structured Day Hypothesis* (*SDH*), children usually show less desirable health behaviors (e.g., screen time, physical activity, eating) during less structured days such as weekends or summer months [21,22], compared to school days. The less structured periods allow for greater behavioral autonomy which leads to increased sedentary behaviors including passive screen time among children [23]. Recently, the *Movement Behaviors Questionnaire* (*MBQ*) has been developed and validated to address this research gap. The MBQ has child-specific (MBQ-C) items that assess passive, interactive, sedentary, and standing screen time on weekdays or weekend days [24,25]. The advent of this novel measure has enabled the field to comprehensively assess and understand screen time as a multidimensional health-related behavior. As a newly developed measure, the open-ended version of the MBQ-C has demonstrated acceptable reliability, with significant positive correlations observed between MBQ-C reported screen time and 24-hour diary ($\rho = 0.44$–$0.86$). However, the MBQ has not been adopted in published research to comprehensively measure children's screen time and its relationship with weight status.

Furthermore, when examining the relationship between screen time and weight status, it is important to consider demographic factors such as age, sex, race, and socioeconomic status (SES). Although prior research has found higher screen time

among older, racial/ethnic minority children from lower SES households [26,27], there is no evidence on how weight status interacts with demographic factors when illustrating screen time patterns. Examining these demographic moderators is important for designing and delivering screen time interventions tailored for specific populations.

Therefore, this study aims to (1) comprehensively examine the association between screen time (various types on weekdays and weekend days) and weight status in a largely disadvantaged child population, and (2) explore whether this relationship would differ across demographic factors. Specifically, the study has two research questions. First, to what extent does screen time or screen time categories differ between OWOB and children with NW in this vulnerable child population? Second, do demographic factors including age, sex, race, and SES (i.e., household income) moderate the relationship between screen time and weight status?

## Methods

### Setting and participants

The data reported in this study stemmed from a larger funded survey study that occurred in Louisiana between October 2023 and August 2025. We initially followed a stratified sampling strategy through community schools (considering urbanicity and area deprivation index) to invite parent proxies of children who were 5–11 years old. This effort led to 543 responses, 311 of which were valid. Nine superintendents, who oversaw forty-one elementary schools, facilitated initial recruitment using stratified sampling. To increase the sample size, we opened the survey to broader regions in the state. Multiple Louisiana Medicaid regional offices facilitated our second recruitment, including the distribution of promotional flyers. The Medicaid regional office staff were encouraged to distribute the flyer by posting hard copies in communal areas of the building and/or by emailing electronic version to Medicaid residents that their office serves. We received 6,830 additional responses, but after applying the initial eligibility criteria and a series of data-quality checks to ensure that responses were legitimate, human-generated responses, only 428 of these additional responses were deemed valid and then included in the analysis. The remaining responses that met the exclusion criteria were removed from analysis. Table 1 outlines these criteria.

**Table 1. Exclusion Criteria of Participants.**

| Exclusion Criteria |
| --- |
| • Incomplete surveys |
| • Insufficient completion duration time (< 15 mins) |
| • Clusters of submissions received with identical timestamps |
| • Extreme body height (e.g., > 6 ft in 2nd grade or younger) |
| • Extreme body weight (e.g., > 200 lbs. or < 20 lbs. in 2nd grade) |
| • Parent proxies of children ≤ 4 or ≥ 11 years old |
| • Grade ineligibility |
| • Inconsistent age–grade alignment |
| • Simultaneous large batches from one location |
| • Non-Louisiana or invalid Zip Code |
| • Duplicate or bounced email addresses |
| • Unusual, suspicious, fabricated email addresses |
| • Excessive daily sleep duration (> 13 hours/day) |
| • Out-of-range weekly bedtime routine values (> 7 nights/week) |
| • Extreme dietary intake values (> 10 fruit/vegetable servings/day) |
| • Incongruent income–resource responses (e.g., household income > $100k but reporting food insecurity or utility shutoffs). |

                                                                    

After applying the exclusion criteria, the final sample included 739 participants. Most surveys were completed during the school semester ($n = 737$, 99.72%), with most responses collected in the early stages of survey dissemination. As shown in Table 2, the sample consisted of more male (58.2%) than female (41.7%) participants balanced by race (White vs. non-White). Most child participants of the parent responders (86.9%) were in higher elementary grades (3rd–5th grade), with a mean age of 9.27 ($SD = 1.49$). A vast majority of families were Medicaid-enrolled (83.9%). Income categories were created for analytic purposes using an approximated tertile distribution with the U.S. median household income (2024 median household as $83,730) used as the reference point [28]. Of all, about one-third of children (32.6%) were deemed OWOB (≥ 85th BMI percentile). This study received ethical approval from the Institutional Review Board of the Louisiana State University (IRBAM-22–0978). Participants provided electronic informed written consent before taking the survey.

## Variables and measures

Screen time measurement utilized eight relevant questions from the MBQ-C open form. These questions consist of four items capturing children's passive screen time and four items capturing interactive screen time. Passive screen time was characterized as time spent watching television programs, videos/internet clips or movies on a television, computer, or portable/mobile device such as iPad, tablet, or smartphone. Interactive screen time was defined as time spent playing games, looking at photos, or video chatting (e.g., FaceTime, Zoom, Skype) on a screen-based device such as a computer or laptop, video game console, iPad, tablet, or smartphone. Parent proxies were prompted to report their children's screen

**Table 2. Demographic characteristics of children.**

| Demographics | Frequency (n) | Percentage (%) |
|---|---|---|
| Sex | | |
| Male | 430 | 58.2 |
| Female | 308 | 41.7 |
| Prefer not to say | 1 | 0.1 |
| Race | | |
| White | 377 | 50.9 |
| Non-White | 362 | 49.1 |
| Grade | | |
| Lower grade (K – 2nd grade) | 271 | 13.1 |
| Higher grade (3rd grade – 5th grade) | 468 | 86.9 |
| Medicaid Enrollment | | |
| Yes | 620 | 83.9 |
| No | 107 | 14.5 |
| Do not know | 12 | 1.6 |
| Household Income | | |
| Low income (~ $40,000) | 249 | 34.7 |
| Middle income (~ $80,000) | 259 | 35.0 |
| High income (above $80,000) | 203 | 27.4 |
| Prefer not to say | 28 | 3.8 |
| Weight Status | | |
| Healthy or underweight | 498 | 67.4 |
| Overweight or Obesity | 241 | 32.6 |
| Total | 739 | 100.0 |

time (in hours and minutes) in the following order using posture-labeled items from the MBQ-C: (1) passive sedentary screen time on a typical weekday, (2) passive standing screen time on a typical weekday, (3) passive sedentary screen time on a typical weekend day, (4) passive standing screen time on a typical weekend day, (5) interactive sedentary screen time on a typical weekday, (6) interactive standing screen time on a typical weekday, (7) interactive sedentary screen time on a typical weekend day, and (8) interactive standing screen time on a typical weekend day. In addition, following the MBQ-C scoring instructions, the total screen time, total sedentary screen time, and the combined sedentary screen time for engagement types were automatically aggregated using Research Electronic Data Capture (REDCap).

### Weight status

We asked parents to report their children's weight, height, age, and sex. They also reported the method of measurement for weight and height (e.g., directly measured, obtained from a recent medical appointment). We subsequently calculated BMI, BMIz, and BMI percentile scores. School-aged children were classified as children with NW or OWOB by 85th BMI-percentile cutoff, based on the CDC BMI categories for children and adolescents aged 2–19 [29]. In this study, NW included underweight and healthy weight, whereas OWOB included all children with BMI percentile 85th or higher. The BMIz and BMI percentiles were calculated using the growth R package for CDC growth charts (*cdcanthro.R*) [30]. This package included the 2022 extended BMI z-score and percentile calculations for children and adolescents with obesity.

### Data collection and processing

Data collection occurred through an online survey administered via the REDCap [31,32]. The survey link and accompanying instructions were distributed to recruit eligible participants. Each participant who completed the survey and met all inclusion criteria received a $20 gift card incentive to compensate for their time and effort. Following the WHO's criteria, we removed biologically implausible observations that were outside ±5 standard deviations from the mean for the BMIz scores [33,34]. We also truncated screen time data points outside the 0–960 min/day range using the Interquartile Range (IQR) rule.

### Data analysis

To address our research question, we conducted the analysis of covariance (ANCOVA) using screen time variables (e.g., passive and interactive screen time on weekdays and weekend days) as outcome variables, weight status as independent variable and sex (Male vs. Female), race (White vs. non-White), household income (tertile scores), and grade as covariates. The data were reviewed for the assumptions of normality and homogeneity of variances. Estimated marginal mean differences (*EMMs*), 95% confidence intervals (CIs), Bonferroni adjusted *p*-values, and effect sizes ($\eta_p^2$) were obtained and reported from this process. We further examined if the two weight status groups would differ in meeting the screen time guidelines: 1 = adherence (≤ 2 hours per day) vs. 0 = non-adherence (> 2 hours per day). Group differences were first examined using a Chi-square test, followed by a logistic regression model with weight status as the predictor and adherence (vs non-adherence) as the outcome. For the logistic regression analysis, we reported the odds ratios (ORs) with 95% CI. These above statistical analyses were conducted in R (version 4.4.1) and RStudio (version 2024.04.2 + 764), with standard modeling and post-estimation packages and SPSS Statistics (version 23) for additional descriptive analysis.

## Results

### Screen time by weight status

As illustrated in Fig 1, about one quarter of the sample (24.9%) met the 2-hour daily screen-time recommendation; however, a significantly lower percentage of children with OWOB (18.6%) met the screen time guideline compared to children with NW (28.0%; $\chi^2$ (1) = 7.12, $p = 0.008$). Logistic regression further verified that children with OWOB had 41% lower odds of meeting the screen time guideline than their NW counterparts (*OR* = 0.59, 95% *CI* [0.39, 0.86], $p = 0.008$).

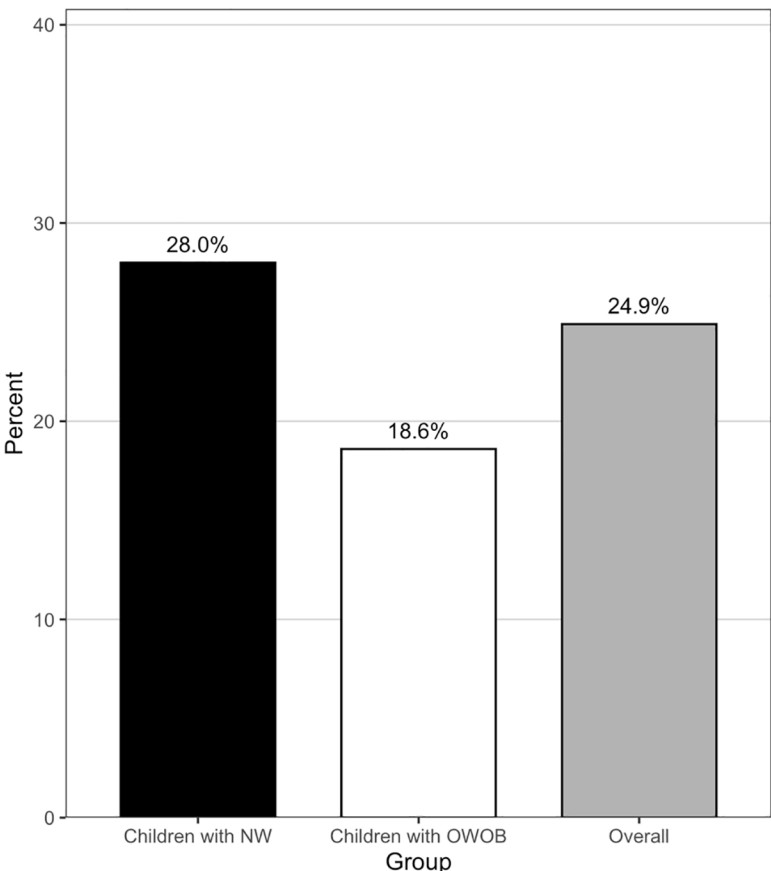

**Fig 1. Percentage of Children Meeting Recommended Daily Screen Time Guidelines (≤ 2 hours).**

Table 3 summarizes children's screen time (by engagement type and posture) by weight status on weekdays and weekend days. Analyses of the total screen time variables indicate that children with OWOB spent substantially longer time on electronic devices than children with NW, with a difference of 69.8 minutes per day for total screen time and 44.6 minutes per day for generally sedentary screen time (95% *CIs*: [45.0, 94.6] and [27.0, 62.1], respectively). Children with OWOB averaged over 6 hours of total screen time per day (364.3 minutes) versus approximately 5 hours among children with NW (294.5 minutes). Of the total screen time, 227.7 minutes and 183.2 minutes were in usually in sedentary posture for the two subgroups, respectively. Meanwhile, screen time was higher on weekends than on weekdays across both weight groups (e.g., passive sedentary screen time: NW group = 90.9 ± 3.2 min on weekdays vs. 135.7 ± 4.6 min on weekends), except for typically standing screen time among children with OWOB. The mean values for passive screen time were consistently higher than those for interactive screen time within the same strata (e.g., passive weekday sedentary vs. interactive weekday sedentary). Finally, the difference in screen time by weight status was significant for all variables on weekdays, indicating that children with OWOB spent more time on various forms of screen time than children with NW ($p < 0.05$). In contrast, several weekend day comparisons were not statistically significant ($p > 0.05$), meaning that weight related difference in screen time were less pronounced during weekends.

Fig 2 illustrates the mean differences of screen time behaviors (i.e., interactive and passive screen time, in posture-related categories labeled as sedentary or standing, on weekdays or weekend days) by weight status (OWOB–NW).

**Table 3. Screen Time (min) by Weight Status across Weekday and Weekend Days.**

| Screen Time Variables | | N | NW | OWOB | $M_{Diff}$ | 95% CI | $\eta_p^2$ |
|---|---|---|---|---|---|---|---|
| | | | Mean ± SE | | | | |
| **Total** | Total | 662 | 294.5 ± 7.2 | 364.3 ± 10.6 | 69.8*** | [45.0, 94.6] | 0.044 |
| | Sedentary | 655 | 183.2 ± 5.1 | 227.7 ± 7.4 | 44.6*** | [27.0, 62.1] | 0.037 |
| **Weekday** | Interactive sedentary | 667 | 65.3 ± 2.6 | 80.9 ± 3.9 | 15.6* | [6.6, 24.7] | 0.017 |
| | Interactive standing | 671 | 41.8 ± 2.2 | 53.9 ± 3.2 | 12.2* | [4.6, 19.7] | 0.015 |
| | Interactive total | 676 | 113.9 ± 4.1 | 141.4 ± 5.9 | 27.5** | [13.6, 41.3] | 0.022 |
| | Passive sedentary | 646 | 90.9 ± 3.2 | 134.5 ± 4.7 | 43.6*** | [32.6, 54.7] | 0.086 |
| | Passive standing | 675 | 52.0 ± 2.8 | 78.8 ± 4.1 | 26.9*** | [17.2, 36.5] | 0.043 |
| | Passive total | 661 | 151.8 ± 4.5 | 215.0 ± 6.6 | 63.2*** | [47.6, 78.7] | 0.089 |
| **Weekend** | Interactive sedentary | 678 | 91.8 ± 3.4 | 102.6 ± 4.8 | 10.8 | [-0.5, 22.2] | 0.005 |
| | Interactive standing | 678 | 49.2 ± 2.6 | 67.2 ± 3.8 | 18.0** | [9.0, 27.0] | 0.023 |
| | Interactive total | 689 | 154.0 ± 4.8 | 166.6 ± 7.0 | 12.6 | [-3.7, 29.0] | 0.003 |
| | Passive sedentary | 681 | 135.7 ± 4.6 | 156.5 ± 6.6 | 20.9 | [5.3, 36.4] | 0.010 |
| | Passive standing | 662 | 55.9 ± 2.8 | 75.2 ± 4.1 | 19.3** | [9.7, 28.9] | 0.023 |
| | Passive total | 673 | 201.0 ± 5.2 | 232.3 ± 7.5 | 31.3** | [13.6, 49.0] | 0.018 |

$p^*$ < 0.05, $p^{**}$ < 0.01, $p^{***}$ < 0.001 (Bonferroni corrected $p$-value). NW: Children with normal weight, OWOB: Children with overweight and obesity.

Screen time difference by engagement type and timing was most evident in passive sedentary screen time on weekdays (43.6 minutes). Sedentary screen time during weekends did not show statistical significance by weight groups. The discrepancy for passive screen time was larger in all postures than that for interactive screen time. These findings suggest that the excessive screen time exposure among children with OWOB were most pronounced for passive screen time, especially on weekdays.

To address the second research question, the two-way ANOVAs revealed no significant interactions of age, sex, race, or SES with weight status for any of the screen time variables ($p > 0.05$). This set of results suggests that the relationship between screen time and weight status remains the same, regardless of these demographic characteristics.

## Discussion

The study aimed to comprehensively examine the association between screen time and weight status and to explore if this relationship would differ by demographic factors in a largely socioeconomically disadvantaged child population. The findings are discussed below.

First, our study was consistent with observations made in prior research that in this population of mostly Medicaid beneficiaries from lower income households, most children engaged in excessive screen time and demonstrated low adherence to the screen time guideline, reflecting the recent trends in media usage among children [14,17]. More importantly, children with OWOB reported higher amount of screen time than children with NW and they were also less likely to meet the sedentary screen time guideline, consistent with prior research [18,35,36]. This observation indicates that tailored interventions are warranted to curb children's screen time, especially for children with overweight or obesity.

Second, our study further revealed a contextual pattern of children's screen time. Overall, children reported larger screen time on weekend days than on weekdays across weight status groups. This supports the structured days hypothesis, suggesting that the structure inherent in weekdays is associated with lower screen time [37]. However, weight-status-based screen time differentials were not consistent. The discrepancy of screen time by engagement type (passive vs. interactive) and day of week (weekday vs. weekend) was greatest on passive screen time during weekdays. Children

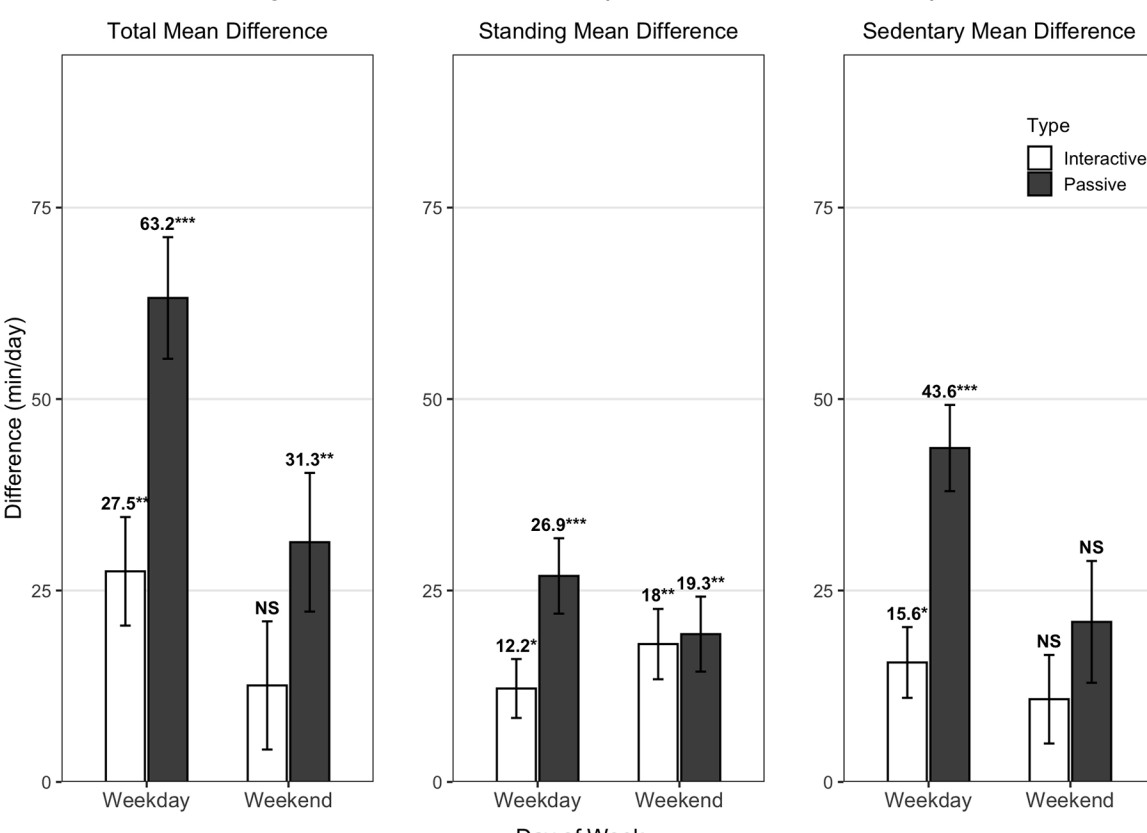

**Fig 2. Differences in Daily Screen Time (OWOB-NW) by Posture, Engagement Type, and Day of Week.**

with OWOB had greater amount of screen time than children with NW on weekdays, whereas on weekends the screen time variables did not show significant difference by weight status. The observed weekday differences by weight status for screen time (e.g., passive screen time) are most likely accrued during after-school hours. Factors such as neighborhood safety, parental supervision, and access to recreational opportunities have been identified to be associated with screen time in prior research [38–40]. On the other hand, the relatively smaller differences observed on weekends may reflect uniformly high screen use across both weight-status groups on these less structured days, where reduced influence of contextual factors (e.g., after-school routines, parental supervision) allows overall screen exposure to rise. This elevated weekend screen time likely reduces between-group variability, attenuating differences between children with OWOB and those with NW. This set of findings suggests that efforts to reduce overall screen time especially for passive screen time among disadvantaged children may be warranted on weekend days. More importantly, future work should purposely look to curtail the screen time during after school hours of the weekdays among children with OWOB.

The results for the second research question further suggest that the relationship between screen time and weight status is stable across demographic characteristics such as age, sex, race, and SES. The absence of significant moderation by demographic variables may reflect several factors. For instance, limited variability or imbalance by demographic characteristics (e.g., over 80% of the sample being Medicaid-enrolled), measurement limitations of screen-time categories, or potential model misspecification may have reduced the ability to detect interaction effects. It is also possible that these demographic factors may not meaningfully moderate or alter the associations.

The strength of this paper is in its contribution to the accumulating evidence that children with overweight or obesity consistently show higher screen time than children with NW. Using a predominantly Medicaid-insured sample from a southern state in United States further expands the relevance of these findings by providing insight into an under-resourced population. Additionally, prior studies have demonstrated associations between weight status and screen time, but this study adds specificity of screen time parent-reported by posture category and engagement types. Presenting these differences with a temporal structure (weekday vs. weekend days) provides an improved understanding of screen time as a multidimensional construct (and its relationship with weight status) from the SDH perspective.

Despite the novel findings, we acknowledge that our measurement of children's screen time and body weight and height were self-reported by parent proxies (primarily mothers), which may have measurement related limitations. Accurately and efficiently measuring children's screen time in a large sample is challenging. Most of the evidence assessing the associations between sedentary behaviors and health outcomes in children and adolescents is cross-sectional, with majority of studies being conducted based on self- or parent-reported measures of sedentary time [41]. Such measures may be subject to reporting bias, recall error, potential misclassification of sedentary versus standing screen time, and misreporting of height and weight, which might have limited our data accuracy and the observed associations. Future studies would benefit from incorporating objective measures of screen-related behaviors (e.g., accelerometry) along with ecological momentary assessments to improve measurement precision and contextual details. The screen-time items from the MBQ-C have demonstrated sound validity and have allowed for comprehensive assessment of screen time by posture, engagement type, and timing [25], which may lead to a paradigm shift in future research on children's screen time, including conceptualization, measurement, and intervention. Yet, it is important to note that the MBQ-C was originally validated among young children from 0 to 5. MBQ-C questions that had content validity for this age range as determined by our team with expertise in developmental psychology, behavioral science, and kinesiology were included to comprehensively measure screen time (engagement type, posture, timing). Investigation for additional validation and reliability of the MBQ-C among older children or adolescent populations is warranted, as their screen time types and patterns may differ by age. We did not find a similar screen-time instrument with our questions of interest that had been validated in our age range, thus we accept this as a limitation of the study. Nonetheless, we believe these are important data to inform the field on screen usage in children prior to their transition to adolescence.

Additionally, "sedentary" and "standing" screen time indicate parent-reported screen use in typically seated or standing contexts rather than objectively measured postures. Therefore, caution is needed when interpreting the results. The observed differences in screen time have relatively small to moderate effect sizes. The practical and public health relevance of these observed patterns should be interpreted with caution. While these differences were relatively small to moderate, these differences bear important implications as the baseline screen time exposure is high.

The findings from this study shed light on future efforts devoted to mitigating screen time. Parents and children are recommended to purposely monitor and curb the amount of passive screen time especially on weekdays among OWOB children. Prior work shows that weekday screen behaviors are more responsive to parental regulation [42], and our results similarly indicate that weekday passive screen time is a meaningful target for intervention, especially because weight-status-based differences were most prominent during weekdays. Consistent with studies demonstrating that parental monitoring and limit-setting reduce excessive or problematic screen use [43], parents may play a key role in shaping healthier weekday routines. Given the greater amount of passive screen time among children with OWOB, these children should be prompted to take frequent breaks from prolonged passive screen time and replace them with physical activity both indoor and outdoor. Parents should provide interventions to establish family rules, goals, and routines to limit excessive screen time. Increasing the proportion of parents who follow the American Academy of Pediatrics recommendations on limiting screen time for children aged 6–17 years is a high priority issue within the Healthy People 2030 objectives, indicating significant public health implications [44]. Furthermore, future studies should also consider incorporating detailed assessments of children's participation in organized and non-organized sports to examine how

these behaviors interact with sex, age, and socioeconomic status in shaping screen-time patterns. Understanding these behavioral and demographic interactions may offer a more comprehensive explanation of variability in screen-time engagement.

## Conclusions

In summary, this study found that screen time distribution was higher among children with OWOB compared to their NW counterpart in a socioeconomically disadvantaged child population, although both groups exhibited excessive amount of screen time daily. Particularly, weekday passive screen time showed the largest difference, suggesting the added urgency for mitigating passive screen time during weekdays among children with OWOB in future interventions. Our study highlighted the importance of considering children's engagement types, parent-reported posture-related categories, and timing of their screen time behaviors distribution that is not possible to capture with a uniform screen-time measure. Future research and practice focused on the conceptualization, measurement, and intervention of children's screen time should consider these dimensions to reflect a comprehensive scope of differential screen time patterns.

## Acknowledgments

The authors would like to thank the participating schools and Medicaid regional offices, and parent participants for their assistance, support, or data contribution.

Disclaimer: The content presented in this paper is the sole responsibility of the authors and does not necessarily represent the official views of the sponsors. The funders had no role in study design, data collection and analysis, decision to publish, or preparation of the manuscript.

## Author contributions

**Conceptualization:** Amanda E. Staiano, Fahui Wang, Stewart Gordon, Senlin Chen.

**Data curation:** Paul Son, Yuxin Nie, Qiaoyin Tan, Peyton Murray, Amanda E. Staiano, Senlin Chen.

**Formal analysis:** Paul Son, Senlin Chen.

**Funding acquisition:** Amanda E. Staiano, Fahui Wang, Stewart Gordon, Senlin Chen.

**Investigation:** Paul Son, Yuxin Nie, Qiaoyin Tan, Senlin Chen.

**Methodology:** Amanda E. Staiano, Fahui Wang, Stewart Gordon, Senlin Chen.

**Project administration:** Senlin Chen.

**Supervision:** Senlin Chen.

**Visualization:** Paul Son.

**Writing – original draft:** Paul Son, Senlin Chen.

**Writing – review & editing:** Paul Son, Yuxin Nie, Qiaoyin Tan, Pengfei Yang, Peyton Murray, Amanda E. Staiano, Fahui Wang, Gang Hu, Stewart Gordon, Senlin Chen.

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
