## [Decision Letter · Decision Letter 0]

12 Jan 2026

PONE-D-25-63306Socioeconomically Disadvantaged Children's Screen Time Behaviors by Weight StatusPLOS One

Dear Dr. Chen,

Thank you for submitting your manuscript to PLOS ONE. After careful consideration, we feel that it has merit but does not fully meet PLOS ONE’s publication criteria as it currently stands. Therefore, we invite you to submit a revised version of the manuscript that addresses the points raised during the review process.

We look forward to receiving your revised manuscript.

Kind regards,

Rajendra Prasad Parajuli, PhD

Academic Editor

PLOS One

Journal Requirements:

This study was supported by the Public University Partnership Program at the Louisiana Department of Health, Bureau of Health Services Financing (LDH-PUPP-AM230210), Louisiana State University Provost’s Fund, Louisiana State University Foundation/Our Lady of the Lake Health.

Dr. Staiano and Hu were partially supported by the grant from the National Institute of Diabetes and Digestive and Kidney Diseases (R01DK141453), the National Institute of General Medical Sciences (U54GM104940).

Reviewers' comments:

Reviewer's Responses to Questions

**Comments to the Author**

1. Is the manuscript technically sound, and do the data support the conclusions?

Reviewer #1: Yes

Reviewer #2: Yes

Reviewer #3: Yes

2. Has the statistical analysis been performed appropriately and rigorously?

Reviewer #1: Yes

Reviewer #2: Yes

Reviewer #3: No

3. Have the authors made all data underlying the findings in their manuscript fully available?

Reviewer #1: No

Reviewer #2: Yes

Reviewer #3: Yes

4. Is the manuscript presented in an intelligible fashion and written in standard English?

Reviewer #1: Yes

Reviewer #2: Yes

Reviewer #3: Yes

5. Review Comments to the Author

Reviewer #1: This manuscript examines differences in screen-time behaviors by weight status in a socioeconomically disadvantaged pediatric population. The topic is relevant, the sample size is adequate, and the use of a multidimensional screen-time instrument is a strength.

The analyses are generally appropriate; however, several issues require clarification or revision before the manuscript can be considered for publication.

First, causal or directional language should be revised throughout the manuscript, as the cross-sectional design only allows for assessment of associations.

Second, the Data Availability Statement does not clearly comply with PLOS ONE policy. The manuscript alternately states that data are available within the manuscript and that datasets require an IRB-approved data-sharing agreement. This inconsistency must be resolved.

Third, although multiple statistically significant findings are reported, many effect sizes are small. The authors should more clearly discuss the practical or public-health relevance of these differences.

Fourth, parts of the discussion and conclusions place emphasis on weekend screen time, despite results indicating that the largest weight-status differences occur for weekday passive screen time. The interpretation and recommendations should be more closely aligned with the results.

Finally, minor editorial issues (typographical errors and repetitive phrasing) should be corrected.

Overall, the study addresses an important question, but the issues noted above should be addressed to improve clarity, transparency, and interpretability.

Reviewer #2: This manuscript as a novel approach examines multidimensional screen time behaviors among children with overweight/obesity (OWOB) compared to their normal-weight peers in a socioeconomically disadvantaged population. So, congratulations to the research team for taking a good initiative study and overall, the study addresses an important and relevant public health issue making meaningful contribution to the literature by moving beyond total screen time to examine engagement type, posture, and weekday versus weekend patterns.

With these, I have few broad queries and suggestions as part of preliminary screening and I hope the research team will address it adequately.

1. Please clarify throughout the discussion and conclusion that the findings represent associations only, given the cross-sectional study design, and avoid any language that may imply causality.

2. Write a few lines about the potential impact of parent-reported screen time and anthropometric data, particularly the possibility of differential reporting, and how this may influence the results.

3. In the findings section, clarify the interpretation of the non-significant moderation effects by demographic variables, including whether limited variability within the sample or statistical power may have contributed to these findings.

4. In findings and discussion section, please clarify the interpretation of the relatively smaller differences in screen time observed on weekends and discuss whether uniformly high weekend screen use across groups may have attenuated between-group differences.

5. In data availability statement, clarify what types of de-identified or aggregated data and/or analytic code may be available upon reasonable request.

6. As part of public health implications, please clarify how the findings particularly the role of weekday passive screen time can inform targeted intervention strategies.

Thank you! Feel free to write back for clarity if confusion arises.

Reviewer #3: Overall comments:

This cross-sectional online survey study reports differences in standing vs active screentime in OW and HW children from a predominantly lower income population. The results show that OW children engaged in more screentime and less active media time and overall have more screentime than HW children. As reported, the results are not new, although the use of a novel questionnaire provides additional nuance to the relationship between screentime and weight status. The authors do a good job excluding potentially problematic data and correctly note study limitations. However, there was no attention to the time of year (summer vs. school) that parents completed the survey, which seems to be an important factor given the Structured Day Hypothesis that the study was based on. This and some additional considerations are below.

Introduction:

The introduction was well-written, clearly pointed out the gap in knowledge, and was theoretically based. However, I do have some suggestions about the title of the paper

and use of the term “socioeconomically disadvantaged” – The paper is primarily about the association between weight status and movement behaviors in a population that is predominantly, but not entirely socioeconomically disadvantaged. In fact, 27.4% of the families are categorized as high income (above the median income). I think the title overemphasizes the demographic characteristics and is not informative enough about the weight status differences (which were the primary finding).

Methods:

1. What was the justification for the age range used in the study? What age range has the Movement Behavior Questionnaire been previously used in?

2. It’s unclear why the question about “looking at photos” was included as part of the interactive screentime as this seems like a passive behavior. Can the authors justify this?

3. Lines 130-135 – how were parents provided with instructions or definitions for each of these categories?

4. How was the sample size determined? Was any power analysis performed?

5. It is unclear from the sampling procedures if the time of year the survey was completed was corrected for in statistical analysis, or if only school time assessments were analyzed. According to the Structured Day Hypothesis, children’s screentime would vary depending on the time of year.

Results:

1. The authors report that ~25% of children met daily screentime requirements, but it’s unclear what recommendations were used for assessing daily screentime. Can this be added? Can the authors cite the specific guidelines that were used here?

2. Line 202 – the authors report on the number of minutes children were engaged in media in the “sedentary posture” however, they did not actually assess children’s physical activity while they were watching media. The assumption is that children were sedentary while engaged in certain types of media, but we cannot verify this assumption. For this reason, the language on how media engagement is described should be tempered accordingly. The same can also be said about “standing” screentime. As activity was not objectively assessed, it’s unclear how parents reported minutes activities that were primary standing vs sedentary.

3. The authors reported no associations between demographic characteristics and movement / screentime, but I’m wondering if they looked at potential interactions between sex and age? For example, do weekend / weekday differences in weight related differences become more robust as children get older and drop out of sporting activities? Is this relationship exacerbated in girls relative to boys? These additional analyses may off the paper additional insight beyond what is largely already known (i.e., OW/OB demonstrate higher sedentary time and greater screen time).

Discussion:

There is no attention in the discussion to the potential effects that involvement in team sports and its associations with sex, age, and SES might have on the reported results.

6. PLOS authors have the option to publish the peer review history of their article (what does this mean?). If published, this will include your full peer review and any attached files.

Reviewer #1: No

Reviewer #2: No

Reviewer #3: **Yes:** Kathleen L. Keller

---

## [Author Response · Author response to Decision Letter 1]

12 Feb 2026

Our point by point responses to the comments are attached and should be available within the overall PDF file. Thank you!

---

## [Decision Letter · Decision Letter 1]

17 Mar 2026

Variations in Children's Screen Time Behaviors by Weight Status among a Mostly Disadvantaged Population

PONE-D-25-63306R1

Dear Dr. Chen,

We’re pleased to inform you that your manuscript has been judged scientifically suitable for publication and will be formally accepted for publication once it meets all outstanding technical requirements.

Kind regards,

Rajendra Prasad Parajuli, PhD

Academic Editor

PLOS One

Additional Editor Comments (optional):

Some typological issues remains: For Example, starting sentense with digit is not common practice, I asked authors to check few issues like..

disadvantaged population. 739 parent proxies of children aged Seven Hundred Thirty nine...

The findings are discussed below....May be not needed..Give a Diligent Read

Reviewers' comments:

Reviewer's Responses to Questions

**Comments to the Author**

1. If the authors have adequately addressed your comments raised in a previous round of review and you feel that this manuscript is now acceptable for publication, you may indicate that here to bypass the “Comments to the Author” section, enter your conflict of interest statement in the “Confidential to Editor” section, and submit your "Accept" recommendation.

Reviewer #1: All comments have been addressed

Reviewer #2: All comments have been addressed

2. Is the manuscript technically sound, and do the data support the conclusions?

Reviewer #1: Yes

Reviewer #2: Yes

3. Has the statistical analysis been performed appropriately and rigorously?

Reviewer #1: Yes

Reviewer #2: Yes

4. Have the authors made all data underlying the findings in their manuscript fully available?

Reviewer #1: Yes

Reviewer #2: Yes

5. Is the manuscript presented in an intelligible fashion and written in standard English?

Reviewer #1: Yes

Reviewer #2: Yes

6. Review Comments to the Author

Reviewer #1: No significant comments.It is a very well written and statistically strong article.I wish the author all the best.

Reviewer #2: (No Response)

7. PLOS authors have the option to publish the peer review history of their article (what does this mean?). If published, this will include your full peer review and any attached files.

Reviewer #1: **Yes:** Anjali Parajuli

Reviewer #2: No

---

## [Editor Report · Acceptance letter]

PONE-D-25-63306R1

PLOS One

Dear Dr. Chen,

I'm pleased to inform you that your manuscript has been deemed suitable for publication in PLOS One. Congratulations! Your manuscript is now being handed over to our production team.

Kind regards,

on behalf of

Dr. Rajendra Prasad Parajuli

Academic Editor

PLOS One